# Reusable Face Masks as Alternative for Disposable Medical Masks: Factors that Affect their Wear-Comfort

**DOI:** 10.3390/ijerph17186623

**Published:** 2020-09-11

**Authors:** Ka-Po Lee, Joanne Yip, Chi-Wai Kan, Jia-Chi Chiou, Ka-Fu Yung

**Affiliations:** 1Institute of Textiles and Clothing, The Hong Kong Polytechnic University, Hong Kong, China; maple.lee@connect.polyu.hk (K.-P.L.); kan.chi.wai@polyu.edu.hk (C.-W.K.); 2Department of Applied Biology and Chemical Technology, The Hong Kong Polytechnic University, Hong Kong, China; jiachi.amber.chiou@polyu.edu.hk (J.-C.C.); kf.yung@polyu.edu.hk (K.-F.Y.)

**Keywords:** COVID−19, face mask, mask comfortability, breathability, fabric mask, reusable mask

## Abstract

The coronavirus outbreak that commenced at the end of 2019 has led to a dramatic increase in the demand for face masks. In countries that are experiencing a shortage of face masks as a result of panic buying or inadequate supply, reusable fabric masks have become a popular option, because they are often considered more cost-effective and environmentally friendly than disposable medical masks. Nevertheless, there remains a significant variation in the quality and performance of existing face masks; not all are simultaneously able to provide protection against the extremely contagious virus and be comfortable to wear. This study aims to examine the influential factors that affect the comfort of reusable face masks, but not to assess the antimicrobial or antiviral potential. Seven types of masks were selected in this study and subjected to air and water vapor permeability testing, thermal conductivity testing and a wear trial. The results indicate that washable face masks made of thin layers of knitted fabric with low density and a permeable filter are more breathable. Additionally, masks that contain sufficient highly thermally conductive materials and have good water vapor permeability are often more comfortable to wear as they can transfer heat and moisture from the body quickly, and thus do not easily dampen and deteriorate.

## 1. Introduction

The COVID−19 pandemic has prompted panic buying and high levels of demand for personal protective equipment (PPE) in many countries, leading to a global shortage of equipment, especially face masks [1,2,3,4]. In Hong Kong, numerous local news outlets, such as the South China Morning Post, have reported that even hospitals and clinics lack an adequate supply of face masks despite the fact that the government has repeatedly pledged that there is enough for medical workers [5]. In situations in which face masks are becoming increasingly scarce, many individuals rush to purchase as many as possible, often overlooking the importance of filtration effectiveness and comfort [6].

Disposable medical masks are the most common type of face masks used to prevent respiratory infections. However, they may allow air and micro-organism leakages or cause choking sensations from feelings of suffocation [7,8]. They are also not the most environmentally friendly and cost-effective option under the current situation. Many manufacturers and investors have, therefore, taken the opportunity to develop reusable and washable face masks to meet market needs. While many newly developed washable masks have been launched in the market, their efficacy is ambiguous. Consequently, it is important to understand the quality of the masks in terms of their effectiveness and comfort.

Permeability is an indicator of the “breathability” of different types of textile materials [9]. It can also be considered one of the most significant attributes used to determine textile comfort [10]. The level of comfort that the textile offers is strongly related to the thermo-regulation and moisture transportation of the material, which means that masks that can quickly transfer heat and moisture away from the face are considered to provide higher levels of wear comfort [11,12]. Because breathing creates a micro-climate inside the mask, wherein the exhaled air and water vapor create their own temperature and humidity levels, excessively high temperature and humidity levels within the mask area may cause a very moist environment and difficulties in breathing. In contrast, masks with a higher thermal conductivity can transfer heat away faster and are able to maintain a lower in-mask temperature. Otherwise, the hot air may accelerate perspiration, and produce sweat that may eventually soak the mask, causing it to cling to the face, and thus increasing discomfort. Moreover, the goodness of fit not only affects the comfort of a mask, but also its filtration effectiveness. An oversized mask, for example, may permit gaps along the side of the face of the wearer, which enables air to infiltrate the mask and therefore virus particles to access the respiratory system, and as such, effectively defeat the initial purpose of wearing a mask.

Many scientists have studied the filtration efficiency of medical masks and N95 respirators. For example, Palen and Felix [13] compared the effectiveness of surgical masks and N95 respirators in preventing influenza, whereas Suen et al. [14] evaluated the reliability of N95 respirators while performing nursing duties. Both Rosenstock [15] and Lam et al. [12] examined the efficacy of face masks and agreed that mask-fit has a dominant role in preventing diseases because it affects the leakage of the seal around the face. However, few studies have focused on reusable and washable fabric masks, and specifically their comfort, which is one of the most important elements that affects wearer compliance. Therefore, this paper focuses on evaluating the comfort of reusable masks and suggesting the influential parameters that might affect their performance. Comfort, here, refers to the breathability, thermal regulation, and goodness of fit. Details of the mask thickness, air and water vapor permeabilities, thermal conductivity and a wear trial are provided and discussed to examine the influential factors that affect mask comfortability.

This study is not aimed to assess the antimicrobial or antiviral potential of the different masks, it evaluates the factors affecting the mask comfort instead. Additionally, the testing performances and impacting factors of seven types of different masks are also compared and discussed in this paper. Section 2 provides the features of these selected masks as stated by their manufacturers.

## 2. Materials and Methods

### 2.1. Materials

Figure 1 shows the images of the samples that were selected for this study.

#### 2.1.1. Sample A—Disposable Surgical Mask

Sample A is a disposable surgical mask produced by HavePur and used as the reference for comparison with the other reusable and washable fabric masks. This surgical mask contains three layers of non-woven fabrics: the outer layer repels droplets; the middle layer filters >95% viruses and > 98% bacteria and particles; and the inner layer provides wear comfort.

#### 2.1.2. Sample B—HODO Mask

Sample B is the HODO mask created by the HODO Sport. According to its official website, the mask received the certificate of FDA Registration on Medical Devices Product, and is capable of blocking 95% of bacteria and 80% of particles [16,17]. This mask can also be used for a total of 200 h if washed properly. Both the outer and inner layers of the mask are made of cotton woven fabric. The filter is composed of multiple layers, which include a nano-fiber micro-porous filter and two polypropylene non-woven films. The company further claims that the mask filter can enhance filtration efficacy while maintaining breathability.

#### 2.1.3. Sample C—M-Chitosan Mask

Sample C, the M-Chitosan, underwent AATCC 100 testing and was shown to inhibit 99.9% of bacterial growth even after 60 machine washes [18]. The material of this mask contains chitosan, which is a natural polysaccharide usually derived from shrimp shells. Chitosan has positively charged molecules that destroy any negatively charged bacteria. The mask consists of 3 layers: the top layer is made of a high density knitted fabric with a smooth surface, which the manufacturer claims to provide water repellency for up to 60 washing cycles; the middle layer acts as the interlining; the inner layer has a lower density knitted structure with a film of chitosan.

#### 2.1.4. Sample D—Accapi Far-Infrared Energy Mask

Sample D, the Accapi Far-infrared energy mask, is made of conventional yarn and incorporates small quantities of metal ions, including titanium, platinum, and aluminum [11]. The mask meets CE Medical Class 1 certification, which means that it is able to inhibit droplets and can emit 4 to 14 microns of far-infrared energy. The company further claims that the mask is both anti-bacterial and capable of deodorizing. The efficacy of the changeable filter is also claimed to be at the same level as that of an N95 mask, which can block 95% of particles that are 0.3 microns in size [15,19]. Therefore, Sample D will have a much longer lifespan than Sample A by merely replacing the filter.

#### 2.1.5. Sample E—Copper Line Mask

Similar to Sample D, Sample E, the Copper Line Mask, is also equipped with anti-bacterial and deodorant functions. According to the official website of Copper Mask and the Hong Kong Copper Line Mask distributer’s Facebook page [20,21], the masks are made of polyester, copper, and polyurethane. Additionally, the use of high-tech fabric with ionized copper yarn can also eliminate 50% of bacteria within 10 min, and over 99.5% in an hour. Moreover, the deodorization effect can reach up to 80% within 30 min, while the anti-bacterial function is shown to remain unaffected after 30 washes. The TTRI test results show that the filtration efficiencies against viruses and particles are up to 99.9% and 99.6%, respectively, if an additional filter is inserted into this three-layer knitted mask [22].

#### 2.1.6. Sample F—PU30 Mask

Unlike the masks discussed above, Sample E, the PU30 mask, is specifically designed to protect the wearer against COVID−19 and the influenza virus (H1N1) [23]. The mask contains three layers: the outer layer is a 100% cotton woven fabric coated with a water repellent cationic antiseptic coating; the middle layer is a polytetrafluoroethylene (PTFE) nanofiber membrane filter; the inner layer is cotton spandex woven fabric to provide extra wear comfort. The modified PU30 mask underwent the ASTM F2100 test, which shows that it can achieve >95% particle, bacterial, and viral filtration efficiency [24].

#### 2.1.7. Sample G—CU Mask

Sample G, the CU Mask, meets ASTM F2100 Level 1 standard, which means that its Particle Filtration Efficiency (PFE) and Bacterial Filtration Efficiency (BFE) exceed 95%, and synthetic blood resistance is more than 80 mmHg [25,26]. The developers also claim that this mask can maintain its functions for up to 60 washes. The mask consists of 6 layers: the outer layer is made with natural fibers and copper, which has the ability to immobilize bacteria, common viruses and other harmful substances [25], while the inner layer is a low density woven fabric. The filter includes one anti-microbial, one filtration and two protective layers. The official website notes that the mask can be used repeatedly by replacing the filter even after 60 washes.

### 2.2. Methods

All samples were conditioned in an atmosphere of 21 °C and 65% humidity for 48 h before testing was carried out.

The outermost layers of the masks were observed under an optical microscope to evaluate their fabric structure and density. The thickness of the masks was also measured in accordance with ASTM D1777. The masks with or without filters were placed onto the base of the thickness gauge. A weighted presser foot of 4.14 kPa was then lowered. The distance between the base and presser foot was calculated and taken as the thickness of the mask. The air permeability of the masks was measured in accordance with ASTM D737. The test samples were placed on the head of the KES-F8 Air Permeability Tester, after which the air was released and passed through the fabric in the perpendicular direction. The airflow was then adjusted to acquire the prescribed air pressure differential between the face and back surface of the mask. The rate of the airflow was then used to determine the air permeability of the mask. Next, a water vapor permeability test was conducted with reference to ASTM E96. The masks were cut and sealed onto the open mouth of a container that contained distilled water. They were then subjected to a controlled temperature (32 °C) and humidity (50%) for 8 h. The rate of water vapor transmission was determined by measuring the weight change of the water at 30-min intervals. Finally, the thermal conductivity test was conducted using a Thermos LABO II. A water-box and BT-box were prepared and set to 25 °C and 35 °C respectively. The masks were then placed in between these two boxes to obtain the data until both boxes reached the same temperature.

A 25-year-old subject with a normal BMI underwent a wear trial with the masks. To simulate an outdoor environment, the temperature and humidity of the room were adjusted to 29 °C and 58% respectively. During the wear trial, the subject was required to sit on a chair, with a temperature logger placed underneath her nose to record the temperature and the humidity in the micro-climate, namely, that inside the masks, for 10 min. The temperature of the face for each mask was also recorded before and after they were worn using an infra-red thermometer.

## 3. Results

### 3.1. Fabric Specifications

Table 1 lists the mask specifications, which include the material used, fabric structure and density per inch of the fabric. Among the washable face masks, only two use woven fabrics as their outer layer, namely Samples B and F. The remainder use knitted fabrics, which are lower in density than the woven fabrics. In total, three masks contain metal in their composition, namely Samples D, E and G.

### 3.2. Fabric Thickness and Air Permeability

Table 2 shows that all of the washable masks with a filter are thicker than the surgical mask, with mean thicknesses of 2.08 mm and 1.24 mm, respectively. Accordingly, the comparatively thinner composition of the surgical mask aligns with its ability to demonstrate a lower air resistance overall. Although thicker masks are expected to have a higher air resistance than thinner masks, the results show a weak correlation between thickness and air resistance (R^2^ = 0.45) (Figure 2). Among the selected masks, the thinnest and thickest masks are Sample G without a filter (0.98 mm) and Sample E (3.22 mm), with an air resistance of 0.06 and 1.56 kPa s/m respectively.

### 3.3. Water Vapor Permeability

The washable masks have higher water vapor permeability than the surgical mask, with mean rates of water vapor transmission of negative 0.40 and 0.30, respectively. Figure 3 shows that Samples C and G have the lowest and highest water transmission rates, which are negative 0.25 and 0.65, respectively. A steeper plotted slope means that the mask can more rapidly transfer moisture and water vapor to the environment, which greatly reduces the in-mask humidity level.

### 3.4. Thermal Conductivity

Four out of the six washable masks showed a higher thermal conductivity compared to the surgical mask. Figure 4 shows the results of the thermal conductivities of the masks. Samples D and E have the lowest and highest thermal conductivities, which correspond to 0.0004018 and 0.0006955W/cm ℃., respectively. When the filter was removed, the thermal conductivity of Sample E was nearly doubled to 0.0012134 W/cm ℃.

### 3.5. Correlation Between Different Test Results

Table 3 shows the correlation between the different test results. As seen, mask thickness has the highest correlation with air resistance and thermal conductivity, with R^2^ values of 0.45 and 0.39, respectively.

### 3.6. Wear Trial

Figure 5 shows the results of the wear trial. With the lowest temperature difference of the face (0.5 °C), the lowest temperature inside the mask (32.5 °C), and relatively low humidity inside the mask (85.5%), Sample D appears to be the most comfortable mask to wear. On the contrary, Sample F appears to be the least comfortable, since it recorded the highest temperature difference of the face (1.1 °C), in addition to a relatively high temperature (34.2 °C) and level of humidity inside the mask (87%).

In terms of goodness of fit, the pleated masks appeared to show a better fit than the 3D masks. With reference to Figure 6 and Table 4, the pleated masks are equipped with a bendable wire, which keeps the masks in place on the face without slipping. The strings around the ears can also be easily adjusted so that the masks can sufficiently cover the face and chin. On the contrary, the 3D masks do not appear to accommodate the contours of the face of the subject, which results in seal leakage. For example, Samples C, D, and G are too large for the subject (Figure 6c–e), so there are gaps around the area of the chin, even if the ear strings are shortened. Moreover, although the nose pad of Sample E is used to prevent leakage around the area of the nose, the bottom of the mask does not entirely wrap around the chin. This causes the mask to easily shift, thus, increasing the potential for leakage.

## 4. Discussion

This paper focus on evaluating the influential factors of mask wear-comfort, so antimicrobial or antiviral potential is not assessed.

### 4.1. Possible Factors that Affect Breathability

Breathability indicates the degree to which a fabric is actively ventilated [9]. To provide more breathability, materials need to easily transfer air and diffuse water vapor into the environment, while preventing the penetration of water droplets into their fibers. This ensures their ability to maintain a dry and comfortable feeling inside the mask. It is extremely important for masks to be breathable; the inability of still air, water vapor, and heat to be effectively transferred will change the micro-climate in the masks, thus resulting in, for example, increased humidity [9]. Excessive heat and trapped water vapor may cause a clammy and damp feeling inside the mask, while further reducing the filtration efficiency and compliance of the wearer.

#### 4.1.1. Thickness

Thinner masks may be commonly perceived by the public to be more breathable than thicker masks. According to the results, however, the correlation between the thickness of the mask and air resistance is not high (R^2^ = 0.45). This means that the air permeability of a mask is not entirely dependent on its thickness. In fact, there could be other attributes that may also affect air permeability, such as the fabric material, density, thickness, and structural properties of the fabric, the diameter of the fibers, and the finishing techniques [9,10].

#### 4.1.2. Fabric Structure: Woven vs. Knitted

Although the difference in the fabric thickness of Sample G without a filter and Sample B is minimal (0.02 mm), there seems to be a significant difference in their air resistance; see Table 2**.** The recorded values are 0.06 and 0.28 kPa s/m respectively. The difference in air resistance is most likely influenced by the fabric structure and density. Whereas Sample G is made of knitted fabric with a lower density per inch (80 × 80), Sample B is made of a plain-woven fabric with a higher density (140 × 80). The loops on knitted fabrics form an “Omega” shape and the knits incorporate interlooping in their fabric structure, and the structure of the woven fabrics use warp yarns that intersect with the weft yarns in a predominantly straight alignment [27,28] (Figure 7). Due to their different structural configurations, woven fabrics can be packed tighter together and are usually associated with higher dimensional stability. Another point to consider is that pores are created on each loop and interlacing point of the knitted fabrics. Although the pore size of the knitted and woven fabrics may appear similar under static conditions, the pore size of the knitted fabrics is easily enlarged when stretched, thus enabling more air and water vapor to pass through. As such, knitted fabrics are usually more breathable than woven fabrics when fitted on dynamic objects under the condition that the same yarn and density are used.

#### 4.1.3. Permeability of Filter

Although some of the changeable filters appear to have excellent filtration efficiency, they are also less breathable. For instance, when testing the masks without a filter, Sample D was thinner and more breathable than Sample E. This supports the general assumption that thinner masks are more breathable. However, when tests were conducted with the filters, the air resistance of Sample D increased by 130%, while that of Sample E increased by only 58%. This shows that filter permeability significantly affects mask breathability.

#### 4.1.4. Goodness of Fit and Climate

The results obtained from the air permeability test are quite different from those from the wear trial. In particular, the masks that demonstrate higher air resistance in the laboratory test were found to be the most breathable in the wear trial. This may have been caused by the goodness of fit, which plays a dominant role in the wear trial [29] but was not taken into consideration in the laboratory experiments. Goodness of fit refers to the ability of the mask to fit the facial contours [30]. Kulichenko [10] noted that the results of laboratory tests conducted in a standard atmosphere can sometimes fail to reflect real conditions when the materials are practically used, since material properties may change under different temperatures. As such, mask comfort may also be greatly affected by the climate in real applications.

According to the laboratory test results listed in Table 2, Sample D is predicted to be the most uncomfortable mask to wear since it has the highest air resistance (1.77 kPa s/m). This is contradicted by the wear trials (Figure 5), which suggest that it is the most comfortable mask to wear since it has the lowest temperature difference of the face (0.5 °C), together with a relatively low in-mask temperature (32.5 °C) and humidity (85.5%). These results could be attributed to the size and design of the mask. When worn, Sample D fails to perfectly accommodate the face of the subject even when the ear strings are shortened. Furthermore, facial movements also cause the filter to shift, thus enabling heat and moisture inside the mask to easily escape through the gaps, which is attributed to the mask being more breathable. On the contrary, Sample F, which is the smallest mask, can perfectly fit the human face, especially with the support of its two bendable wires. Despite its advantages, however, Sample F also shows less breathability. Furthermore, moisture may be trapped inside the mask which would cause a clammy and damp feeling. While loose-fitting masks may appear to be more breathable, they are designed in a way that prevents their ability to perform their intended function of protection from disease.

Goodness of fit appears to be the most important factor that affects mask breathability because it controls the size of the gap between the face and the mask. Compared to large quantities of small fabric pores, a gap, even if small size, is much more capable of permitting greater airflow. Goodness of fit also directly relates to comfort because an overly tight mask may irritate the skin or cause choking sensations, even if the mask is thin. If goodness of fit was to be treated as the primary factor in determining the best mask, Sample B would be selected as the best among all of the masks tested; it fits the face contour perfectly after adjustments of the wire and ear strings, without irritating the skin or causing choking sensations. With the lowest air resistance shown, it is also deemed to be the most breathable mask.

### 4.2. Possible Factors that Affect Thermal Regulation

#### 4.2.1. Thermal Conductivity of Materials Used

Heat transfer involves three main physical mechanisms, namely, radiation, convection and conduction [31]. In particular, thermal conduction is generally believed to be the most significant process for fibrous materials because it occurs when a temperature gradient is present between the material and the environment [31]. Radiation can be neglected when the temperature gradient is small. Similarly, convection can also be disregarded when the friction of the fibers is high because it will be suppressed by the pores. According to Figure 4, Sample E shows the highest thermal conductivity at 0.0006955 and 0.0012134 W/cm °C, regardless of whether a filter is used. Several cases have suggested that there is a positive correlation between air permeability and thermal conductivity, because the entrapped still air appears to provide significantly more insulation than the fibers and thus retains the heat in the in-mask atmosphere [29,32,33]. However, the test results of the masks appear to indicate little correlation between thermal conductivity and air resistance (R^2^ = 0.05), which could be because air permeability is not influenced by the material used for masks, whereas this is not true for thermal conductivity.

Behera and Hari [29] argued that heat flow through materials mainly depends on the fiber content. Typically, conductive materials transfer heat faster than non-conductive materials. This means that blending, mixing, and coating with conductive materials such as metals can potentially improve thermal conductivity. Both Samples D and E consist of metals, but the type and thermal conductivity are different. The former uses small quantities of titanium, platinum and aluminum, while the latter uses copper. At 0 to 100 °C, copper exhibits a much higher thermal conductivity than aluminum, platinum and titanium, which are 401, 237, 71.6, and 21.9 W/(m K), respectively [34,35,36,37]. This suggests that copper transfers heat faster than the other metals, hence, supporting the results that show a higher thermal conductivity in Sample E than Sample D. In contrast, conventional fibers are normally associated with lower thermal conductivity; for example, pure cotton has a thermal conductivity of only 0.04 W/cm °C [38]. Such materials are thermally insulated instead of thermally conductive. Based on these results, it can be presumed that masks with higher thermal conductive metals can transfer heat faster and provide greater thermal comfort.

#### 4.2.2. Permeability of Filters

Non-woven filters can inhibit the efficiency of heat transfer, especially when the filters are made of multiple layers of materials or have a high density. For example, added filters caused the thermal conductivity of Samples D and E to decrease by 33% and 43% respectively. This means that although the changeable filters are equipped with excellent filtration efficiency, there is still room for improvement in thermal conductivity. Filter permeability not only affects mask breathability but also thermal conductivity.

#### 4.2.3. Water Vapor Permeability

Water vapor permeability is an important indicator of thermal comfort [39]. It is generally believed that a higher rate of water vapor transmission can increase wear-comfort because it keeps the mask dry and clean. Perspiration cools down the body, whereby a large volume of water vapor is generated under the condition of a hot climate or heavy work. As such, the materials of the mask should be able to absorb the moisture from the wearer’s skin and transfer the moisture to the environment quickly [40]. Otherwise, high temperatures and humidity in the mask may cause difficulties in breathing. Furthermore, sweat that soaks the mask can also increase the chances of a clammy and uncomfortable feeling. Because most of the washable masks showed higher rates of water vapor transmission than the disposable medical mask, they are presumed to be more comfortable to wear.

According to Figure 3, Samples C and G have the lowest and highest rates of water vapor transmission, respectively. However, it is important to note that because Sample G absorbs an excessive amount of moisture, the mask was soaking wet during testing. This implies that while Sample G has a high moisture absorption rate, its evaporation rate appears to be considerably low. Erdumlu and Saricam [12] suggested that thermal conductivity can be affected by the amount of moisture found in textiles because water is a good conductor for heat transfer. This means that wet textiles may have higher water vapor permeability. It is therefore not surprising that Sample G has a higher rate of water vapor permeability than the other masks. However, because the mask was soaking wet, it cannot be considered to have good wear comfort.

### 4.3. Recommendations

It is important to balance comfort and functionality. This means that a mask should be both breathable and demonstrate sufficient filtration efficacy. Otherwise, the mask could be protective and reliable but lack acceptance and compliance [41,42]. Whereas masks made of thicker yarn are normally capable of showing higher efficiency in removing particulate matter (PM), those made of thinner yarn can provide greater thermal comfort [43]. To create a mask that offers both good protection and high comfort, manufacturers can increase the fabric thickness while minimizing the fiber or yarn thickness. For example, N95 respirators are typically thicker than disposable medical masks, but are breathable and demonstrate higher levels of protection because they are made of finer yarn [44,45,46].

Moreover, it is important to note that the goodness of fit is one of the key parameters for both filtration efficacy and comfort. Usually, a better fit is associated with more tightness and increased protection. However, high levels of pressure on the face might also increase heat, skin irritation or even create pressure marks when the mask is worn for a prolonged period of time [14]. Mask discomfort may eventually reduce the compliance of the wearer [44,47]; some might even refuse to continue wearing a mask. On the contrary, loose-fitting masks are more comfortable to wear, but facial or body movement may cause the mask to shift, and thus enhance the risk of leakage of the seal around the face. Human face contours also vary greatly, which makes it difficult for wearers to select good-fitting masks [48,49]. This means that while a face mask may perfectly fit one’s face, others with different facial features may feel that it is too loose or tight.

Future studies should place greater emphasis on the design features of masks to improve their fit. Some features to consider may include: the shape and material of the ear loops, necessity of bendable wires and nose pads, and the cut. The general belief is that when facial features are taken into account when producing face masks, comfort, compliance and protection [41] are all enhanced.

## 5. Conclusions

In conclusion, while disposable medical masks showed higher air permeability, many of the reusable fabric masks revealed higher thermal conductivity and water vapor permeability. It is also worth noting that pleated masks seem to provide a better fit. However, mask breathability and comfort do not simply depend on one attribute due to their complex composition. Instead, they depend on a range of different parameters, such as fabric thickness, structure and density, fiber content, the permeability of the filter, microclimate, and goodness of fit. The overall belief is that washable masks made of thin and low density knitted fabric together with a good permeability filter will be more breathable. Additionally, masks that consist of sufficient highly thermally conductive materials, such as copper, and good water vapor permeability are assumed to be more comfortable to wear. This is because they allow moisture and heat to be transferred into the environment more easily. However, absorbing too much water vapor without the ability to diffuse the vapor quickly can dampen the mask, which further reduces its wear comfort. Among these factors, goodness of fit has the most significant role when evaluating the quality of masks, because it is the only variable that directly and simultaneously influences mask breathability and effectiveness. It could also be regarded as the key factor in assessing mask comfort, because overly tight masks may irritate the skin and cause choking sensations, even if they are composed of thin materials and characterized by high thermal conductivity and water vapor permeability.

## Figures and Tables

**Figure 1 ijerph-17-06623-f001:**
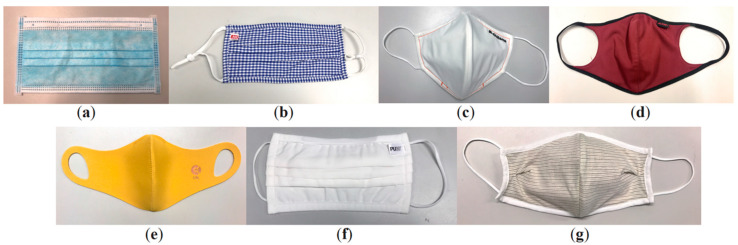
Images of (**a**) Sample A, (**b**) Sample B, (**c**) Sample C, (**d**) Sample D, (**e**) Sample E, (**f**) Sample F, and (**g**) Sample G.

**Figure 2 ijerph-17-06623-f002:**
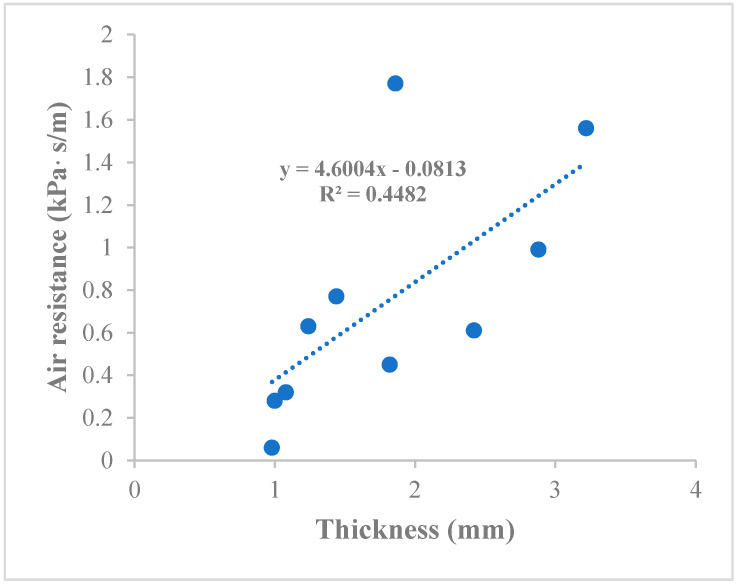
Correlation between thickness and air resistance of mask.

**Figure 3 ijerph-17-06623-f003:**
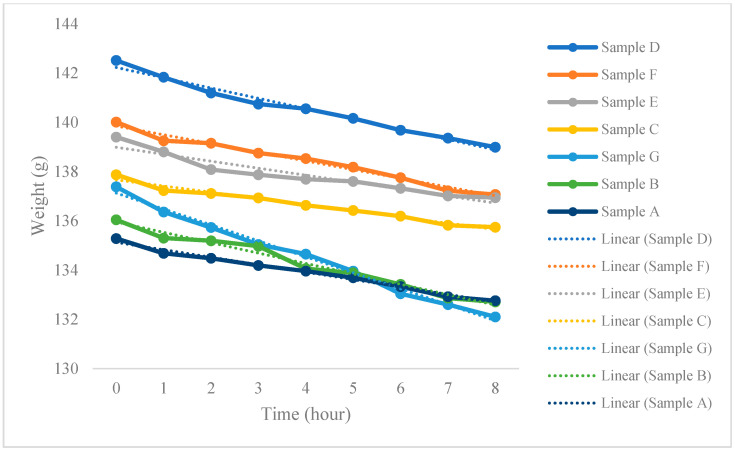
Water vapor transmission of different masks. Note: Water vapor transmission rate (slope): Sample D = −0.4182; Sample F = −0.3537; Sample E = −0.2833; Sample C = −0.2517; Sample G = −0.6477; Sample B = −0.421; Sample A = −0.3027.

**Figure 4 ijerph-17-06623-f004:**
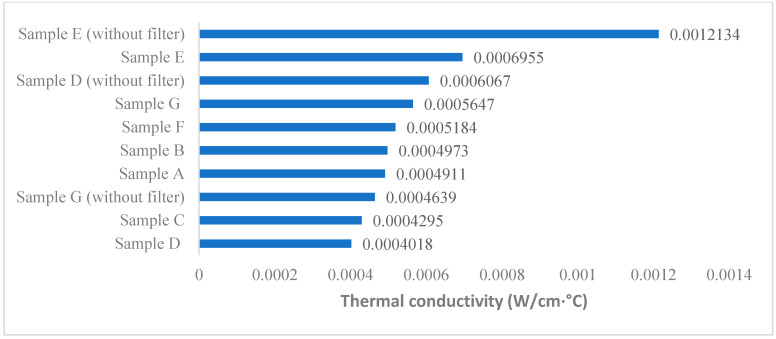
Thermal conductivity of different masks.

**Figure 5 ijerph-17-06623-f005:**
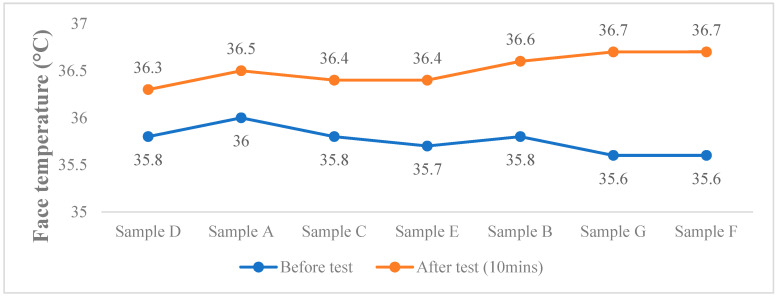
Recorded face temperature difference, in-mask temperature and in-mask humidity of different masks in the wear trial. Note: In-mask temp. (°C): Simple D = 32.5; Simple A = 34.2; Simple C = 33.1; Simple E = 34.5; Sample B = 34.1; Sample G = 34; Sample F = 34.2. In-mask humidity (%): Simple D = 85.5; Simple A = 87; Simple C = 86; Simple E = 86.5; Sample B = 87.5; Sample G = 84.5; Sample F = 87.

**Figure 6 ijerph-17-06623-f006:**
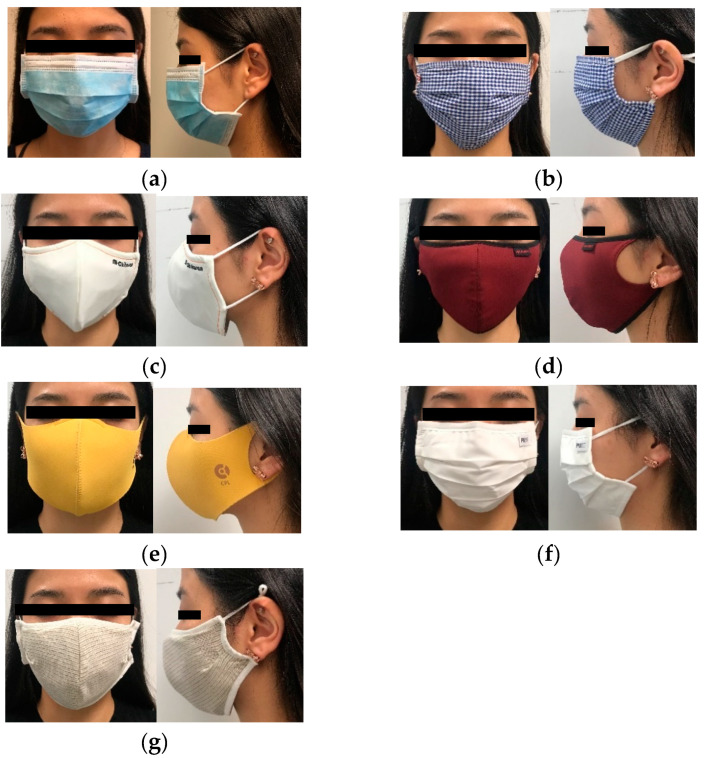
Front and side views of (**a**) Sample A, (**b**) Sample B, (**c**) Sample C, (**d**) Sample D, (**e**) Sample E (**f**) Sample F, and (**g**) Sample G.

**Figure 7 ijerph-17-06623-f007:**
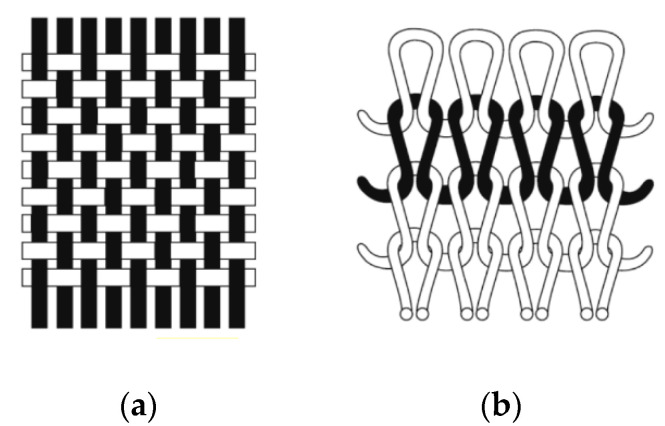
Fabric structure of (**a**) woven and (**b**) knitted fabrics [28].

**Table 1 ijerph-17-06623-t001:** Specifications of different masks.

	Images Under Microscope ^1^	Material Used(Excluding Filter)	Fabric Structure	Density/Inch
Sample A	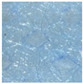	Polypropylene	Non-woven	/
Sample B	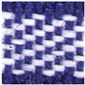	Cotton	Woven	80 × 80
Sample C	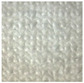	Natural fiber, Chitosan	KnittedInterliningWoven	70 × 140
Sample D	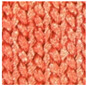	Conventional fiber, Aluminium, Platinum, Titanium	KnittedWoven	50 × 60
Sample E	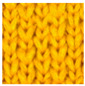	Polyester, Copper, Polyurethane	Knitted	50 × 70
Sample F	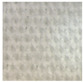	Cotton	WovenNon-wovenKnitted	140 × 80
Sample G	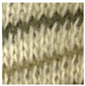	Natural fiber, Copper	KnittedNon-wovenWoven	80 × 80

Image scale: 1/10 × 1/10 inch.

**Table 2 ijerph-17-06623-t002:** Thickness and air permeability of masks.

	Thickness (mm)	Air resistance (kPa.s/m)
Sample A	1.24	0.63
Sample B	1.00	0.28
Sample C	1.82	0.45
Sample D (without filter)	1.44	0.77
Sample D (with filter)	1.86	1.77
Sample E (without filter)	2.88	0.99
Sample E (with filter)	3.22	1.56
Sample F	1.08	0.32
Sample G (without filter)	0.98	0.06
Sample G (with filter)	2.42	0.61

**Table 3 ijerph-17-06623-t003:** Correlation between different test results.

	Correlation between	R^2^ Value
**Breathability**	Thickness and air resistance	0.45
	Water vapor permeability and air resistance ^1^	0.12
	Thermal conductivity and air resistance	0.05
**Thermal regulation**	Thickness and thermal conductivity	0.39
	Water vapor permeability and thermal conductivity ^1^	0.03

^1^ All masks were tested with changeable filters.

**Table 4 ijerph-17-06623-t004:** Size and design characteristics of masks.

	Length (cm)	Width (cm)	Type	Features
Sample A	9.5	17.5	Pleated	Bendable wire
Sample B	8.5	17.5	Pleated	Bendable wire Adjustable ear strings
Sample C	17	19	3D	/
Sample D	14.5	17.5	3D	Changeable filter
Sample E	14	23.5	3D	Changeable filter, Nose pad
Sample F	9	17.5	Pleated	2 Bendable wires
Sample G	14	20.5	3D	Changeable filter

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
