# Peer review of "Reusable Face Masks as Alternative for Disposable Medical Masks: Factors that Affect their Wear-Comfort"

_ijerph, 2020, doi:10.3390/ijerph17186623_

Round 1

Reviewer 1 Report

The manuscript presents results of some physical tests carried out for texitile used as face mask basis.

According to tests and its results:

  • the water permeability and water vapour peremeability are not the same feautures,
  • the relationship between thickness and air permeability or air resistance of textiles is commonly known, depends on many factors, the investigation in this case is not needed,
  • thermal conductivity for all samples is placed on the same level, the analysis of result is wrong, there are no differences, especially for such thin samples - regards thermal properties,
  • wear trial - both measured parameteres are on the same level for all samples, taking under consideration observed differences, which are small and have to be neglacted rather then being detailed analysed,
  • air permeability or air resistance stays in contrast with particle or small droplet filtration. The standards describe max acceptable value of air resistance, connected with filtration.

Are any of presented face masks the certificate (US N95 or EN FF2/FF3)? If not, it is wrong to mention them as ones which meet even one of this standards criteria. All of them cause leackages of air breathed by wearer and taken by them - it is clearly visiby in its construction.

The face mask have to be tested with filter.

The properties of the face mask presented in manuscipt coming from their manufacuteres and they are not supported by certificates (maybe without sample A). They should be investigated as general use face masks.

Reviewer 2 Report

The paper draws an interesting comparison between different types of face masks and its topic is quite timely. However, its quality should be improved by improving the scientific aspect of explaining the collected results: 

  1. The commercial names and brands of the tested samples should be provided, if available.
  2. The scientific aspect of the paper should be improved. The current form of the paper only discusses the collected data without sufficient explanation. The assessed parameters such as moisture permeability, and thermal conductivity should be explained based on the materials type of each face masks.
  3. It should be explained exactly which parameter plays the most important role in improving the wearing comfort of users. 
  4. Was there any advantage of using surgical face masks to other tested samples?  

Reviewer 3 Report

In this manuscript, Lee et.al, examined and reported the influential factors that affect the comfortability of reusable face masks, which is very relevant to the current situation of COVID-19 pandemic. My comments and suggestions as follows.

1, In the introduction section more relevant references, should be added.

2, In table 1, authors should mention which polymer used for sample A.

3,  The contents of section 3 was mixing together. It will be more clear to understand if the 'section 3.2 correlation between different test results' could be rearranged to the end of section 3 or after section 3.5.

3, In the conclusion section, readers will get more clarity if authors can mention which masks were best among the samples to correlate with factors that authors have mentioned in this section.

Round 2

Reviewer 1 Report

This manuscript, after essencial improvments can be published. The most important thing is to put the paper subject and meaning clearly for readers - the filtration and safety properties of presented face masks are not the aim of study. Other changes, in scope of resuts and its description and analysis was done properly.